# Effects of Sprint Interval Training at Different Altitudes on Cycling Performance at Sea-Level

**DOI:** 10.3390/sports8110148

**Published:** 2020-11-18

**Authors:** Geoffrey Warnier, Nicolas Benoit, Damien Naslain, Sophie Lambrecht, Marc Francaux, Louise Deldicque

**Affiliations:** 1Institute of Neuroscience, UCLouvain, 1348 Louvain-la-Neuve, Belgium; geoffrey.warnier@uclouvain.be (G.W.); nicolas.benoit@uclouvain.be (N.B.); damien.naslain@uclouvain.be (D.N.); marc.francaux@uclouvain.be (M.F.); 2Department of Physical Medicine and Rehabilitation, Saint-Luc University Hospitals, 1200 Brussels, Belgium; sophie.lambrecht@uclouvain.be

**Keywords:** hypoxia, cycling, lactate threshold, repeated sprint training, time trial, Wingate test

## Abstract

Background: Benefits of sprint interval training performed in hypoxia (SIH) compared to normoxia (SIN) have been assessed by studies mostly conducted around 3000 m of simulated altitude. The present study aims to determine whether SIH at an altitude as high as 4000 m can elicit greater adaptations than the same training at 2000 m, 3000 m or sea-level. Methods: Thirty well-trained endurance male athletes (18–35 years old) participated in a six-week repeated sprint interval training program (30 s all-out sprint, 4 min 30 s recovery; 4–9 repetitions, 2 sessions/week) at sea-level (SL, *n* = 8), 2000 m (F_i_O_2_ 16.7%, *n* = 8), 3000 m (F_i_O_2_ 14.5%, *n* = 7) or 4000 m (F_i_O_2_ 13.0%, *n* = 7). Aerobic and anaerobic exercise components were evaluated by an incremental exercise test, a 600 kJ time trial and a Wingate test before and after the training program. Results: After training, peak power output (PPO) during the incremental exercise test increased (~6%) without differences between groups. The lactate threshold assessed by Dmax increased at 2000 m (+14 ± 12 W) and 4000 m (+12 ± 11 W) but did not change at SL and 3000 m. Mean power during the Wingate test increased at SL, 2000 m and 4000 m, although peak power increased only at 4000 m (+38 ± 38 W). Conclusions: The present study indicates that SIH using 30 s sprints is as efficient as SIN for improving aerobic and anaerobic qualities. Additional benefits such as lactate-related adaptations were found only in SIH and Wingate peak power only increased at 4000 m. This finding is of particular interest for disciplines requiring high power output, such as in very explosive sports.

## 1. Introduction

Hypoxic training has been used to enhance cycling performance at sea-level for decades. Originally, altitude training camps were organized at moderate altitude (1800–2500 m) for two to four weeks, two or three times a year. This method has been defined as the traditional “Live high–Train high”. Since then, various strategies have been proposed such as “Live high–Train low”, “Intermittent hypoxic exposure” (IHE) or “Intermittent hypoxic training” (IHT) [1]. This last method has gained in popularity especially when combined with high intensity interval training (HIIT). Even if the mechanisms by which hypoxic training has additive effects compared to classical training at sea level remain unclear, some evidence indicates that both aerobic and anaerobic components could benefit from IHT [2,3].

In the early 2000s, a new variation of HIIT consisting of the repetition of short (≤30 s) all-out sprints emerged [4,5]. For a much smaller training volume, sprint training has demonstrated to be as effective as continuous endurance training to increase muscle oxidative capacity [4,6,7,8], maximal oxygen uptake (VO_2max_) [4,7] and cycling time trial performance [6,8]. The two major sprint training protocols can be described as follows: repeated sprint training (RST) is characterized by repeated maximal exercise bouts of short duration (≤10 s) interspersed with brief recovery periods (≤60 s or exercise-to-rest ratio <1:4), whereas sprint interval training (SIT) includes longer sprints (usually 30 s) with 2–4 min recovery [9,10]. When performed in hypoxia, these types of training are defined as RST in hypoxia (RSH) or SIT in hypoxia (SIH). The current study will focus on the latter. Although single aerobic and anaerobic performance seem to be improved in a similar way with SIH in comparison to the same training performed in normoxia (SIN), some specificities have been highlighted [11]. SIH enhanced cycling power output at 4 mmol·L^−1^ lactate during an incremental exercise test [12] as well as the ventilatory threshold [13] when no improvement was reported with SIN.

Although altitude training camps and IHE seem to be efficient for enhancing hematocrit, hemoglobin mass and red blood cells, most of the studies investigating IHT do not report any hematological changes [1]. Being involved in numerous physiological processes such as oxidative metabolism or erythropoiesis, iron is essential for endurance athletes whose athletic performance requires a high aerobic capacity [14,15]. The decline of iron storage in the blood is well documented as a result of altitude training camp [15,16], however there is a lack of data regarding the effects of sprint training protocols on ferritin levels.

Short all-out sprint (<45 s) performance does not appear to be impaired by acute hypoxia. Nevertheless, the decrease in oxygen availability is compensated by an increase in the production of ATP by glycolysis [17]. Due to the lower rate of oxygen delivery, it has been hypothesized that SIH induces a higher expression and activity of glycolytic enzymes, which in turn could enhance performance during a Wingate test or a time trial. In line with the latter, six weeks SIH, but not SIT, increased phosphofructokinase (PFK) activity, despite similar improvement in cycling performance in both groups [12]. The authors hypothesized that other adaptations, such as buffer capacity, might be required to detect an additional effect of SIH over SIT on cycling performance [12]. To date, most of the SIH studies have been conducted around 3000 m of simulated altitude [12,13,18,19]. To our knowledge, no study has shown that this altitude is optimal for maximizing the performance improvements induced by SIH programs. The present study aims to determine whether SIH at an altitude as high as 4000 m can elicit a greater physiological stress and therefore larger adaptations than the same training at 2000 m, 3000 m or sea-level.

## 2. Materials and Methods

### 2.1. Participants

A sample size analysis has been performed according to the superiority formula for parallel RCT with continuous variables and the calculator developed by Wang and Ji [20]. To find a 10% difference in the means of peak power output (PPO) between pre- and post-training with an SD corresponding to 7% [18] and a 2% superiority margin, as well as power of 80% and a significance level of 5%, considering 3 treated groups were tested and compared to 1 control group, it was predicted that 28 participants (*n* = 7/group) were needed if a drop-out rate of 3% was taken into account. The superiority margin represents the threshold above which a group was considered as gaining an advantage from one altitude over another. Thirty-one well-trained endurance male athletes (cyclists or triathletes) competing in amateur categories gave their written consent to voluntarily participate in the experiment, which was approved by the ethical committee of the UCLouvain (B403201939034) and conducted in accordance with the Declaration of Helsinski. The participants were recruited based on their age (18–35 years) and their training volume (4 to 10 h/week). Exclusion criteria for participation were smoking, exposure to an altitude above 1500 m during the month before the experiment, and any health risk that could compromise the participant’s safety during training and/or hypoxic exposure. The participants were asked to maintain their usual dietary habits and training load throughout the experiment. They completed a medical survey to ensure they were not taking any medication or supplements that could interfere with the experiment. Only 2 participants reported occasional whey protein consumption, and none took iron supplementation. They were also asked to avoid strenuous workouts the day before the experimental training sessions. The participants recorded all their personal workouts in a Strava diary to allow us to quantify their training volume. Of the thirty-one participants involved, only one had to stop the study—due to a fall that occurred during a personal training session.

### 2.2. Experimental Protocol

The study was conducted between September 2019 and January 2020 in 2 experimental periods. During each period, 3 phases were organized: 2 weeks of pre-testing, 6 weeks of training and 2 weeks of post-testing. All pre- and post-tests were performed at sea-level. All cycling tests and training sessions were performed using a cycle ergometer (Cyclus II; RBM Electronics, Leipzig, Germany) and the participant’s personal bike. The pre-testing period consisted of 5 visits to the laboratory, which can briefly be summarized as follows. First visit (day 1): height, weight and hemoglobin mass measurement, blood sampling. Second visit (day 3): incremental test and familiarization with the Wingate test. Third visit (day 4): familiarization with the 600 kJ time trial. Fourth visit (day 8): Wingate test. Fifth visit (day 10): 600 kJ time trial. After the pre-test period, the participants were semi-randomly assigned to one of the 4 experimental groups based on the PPO measured during the incremental exercise test to obtain similar values of fitness level, age, and body weight (Table 1). The 4 groups participated in a 6-week supervised cycling training program involving SIT at different altitudes: sea-level (SL, F_i_O_2_ 20.9%, *n* = 8), 2000 m (F_i_O_2_ 16.7%, *n* = 8), 3000 m (F_i_O_2_ 14.5%, *n* = 7) and 4000 m (F_i_O_2_ 13.0%, *n* = 7).

Values are means ± SD for sea-level group (F_i_O_2_ 20.9%,) and altitude groups (2000 m, F_i_O_2_ 16.7%; 3000 m, F_i_O_2_ 14.5%; 4000 m, F_i_O_2_ 13.0%).

Hypoxia was achieved following nitrogen injection into a confined room to dilute oxygen content in the air (High altitude system B-Cat, Tiel, The Netherlands). After the training intervention, all participants participated to the post-test period, which was identical to the pre-test without the familiarization sessions. Blood sampling was performed 3 days after the last training session and the first cycling test was performed 5 days after the last session to allow a complete recovery.

### 2.3. Training Sessions

All participants were blind to the conditions until the end of the experiment. They participated in a supervised SIT session twice a week for 6 weeks based on methods from Puype et al. [12]. The sessions started with a 10 min warm-up at ~150 W (80–90 revolutions per minute (RPM), 17.5 Nm) and two 5 s sprints. Thereafter, the participants had to repeat a series of 30 s sprints with a 4 min 30 s-recovery at ~75 W (80–90 RPM, 8.75 Nm). The first and last sprint of each session were set as a Wingate test (see below). The other sprints were set at 80% of the Wingate load. The participants were asked to ‘go all-out’ on all sprints and try to produce the highest possible power output. The number of sprints per session was progressively increased from 4 in week 1 to 9 in week 6.

### 2.4. Blood Analyses

Hematocrit (Hct) and hemoglobin concentrations ([Hb]) were measured immediately after blood drawing from an antecubital vein of the forearm using an automated device (ABX Micros 60, Axonlab, Dättwil, Switzerland). EDTA tubes were then centrifuged for 15 min at 2000× *g* at 4 °C and plasma fraction was collected and stored at −20 °C. Plasma ferritin concentrations were also measured using an automated analyzer (Pentra C 200, Horiba medical, Axonlab, Dättwil, Switzerland). Blood, plasma and red blood cell volumes (BV, PV and RBCV) were calculated based on hematocrit, hemoglobin concentrations and hemoglobin mass (see section below) [21,22]:RBCV = Hb_mass_/MCHC × 100(1)
BV = RBCV × (100/Hct)(2)
PV = BV – RBCV(3)
where MCHC is the mean corpuscular hemoglobin concentration (([Hb]/Hct) × 100), and Hb_mass_ is the hemoglobin mass.

### 2.5. Hemoglobin Mass Measurement

Hemoglobin mass was measured using a modified version of the optimized carbon monoxide (CO) rebreathing method developed by Schmidt and Prommer [23] and detailed in Meurrens et al. [24]. The reliability of the method is characterized by a typical error of ~1.4% [22,23,25] and the validity of the procedure has been confirmed after a close agreement between the measured and calculated losses of total Hb_mass_ via 550 mL phlebotomy (mean error of 9 g) was found [23].

### 2.6. Incremental Exercise Test

A maximal incremental exercise test was performed to assess aerobic fitness, as determined by VO_2max_ and PPO. The starting load was set at 110 W and was incremented by 40 W every 3 min until exhaustion. Heart rate (HR) (Polar Team System 2; Polar Electro, Kempele, Finland) and respiratory exchanges (Ergocard Clinical, Medisoft, Sorinnes, Belgium) were continuously monitored while a capillary blood sample was collected from the right earlobe in the last 15 s of each stage for the determination of blood lactate concentrations (Lactate Pro, Arkray, Japan). Based on the lactate curve, the power produced at Dmax was calculated according to Cheng et al. (1992) [26]. Due to technical issues, VO_2max_ results cannot be presented.

### 2.7. Wingate Test

The Wingate test consisted of a 30 s maximal cycling exercise with a resistance set to 7.5% of the participant’s body weight (0.075 kg·kg bw^−1^). The test was preceded by a 10 min warm-up without resistance including two 5 s sprints at a test load. The participants were asked to try and generate the highest power possible throughout the test and were strongly encouraged during the 30 s. The following variables were subsequently calculated: peak power, mean power for the 30 s, and the fatigue index representing the decrease in power during the test (W·s^−1^). Blood lactate concentrations were measured 3 min and 5 min post-Wingate (Lactate Pro, Arkray, Japan).

### 2.8. 600 kJ Time Trial (TT)

To simulate a field performance, the participants performed a 600 kJ TT. After a 10 min warm-up at 100 W, the participants had to reach 600 kJ as quickly as possible. The only information given to the participants was the work progression in kJ and the RPM.

### 2.9. Statistics

All values are expressed as the means ± SD. Repeated-measures ANOVA were performed on raw values considering the condition (SL, 2000 m, 3000 m or 4000 m) as the inter-group factor, and time (pre-training and post-training) as an intra-group factor. Tukey post hoc analyses were performed when indicated. Pre- and post-training measurements for each group were compared using a two-tailed paired Student’s *t*-test with 95% confidence interval (CI). All analyses resulting in *p* < 0.05 were considered to be statistically significant, while *p*-values between 0.05 and 0.10 were described as tendencies. The above statistical analyses were performed with the Statistical Package for the Social Sciences (SPSS v.25.0; IBM, Armonk, NY, USA). Effect size (ES) was determined using Cohen’s d developed by Cohen (1988) [27] and discussed in Lakens (2013) [28]. The magnitude of the ES was classified as huge (>2.0), very large (1.19–2.0), large (0.80–1.19), medium (0.50–0.79), small (0.20–0.49) or very small (<0.19) [29].

## 3. Results

### 3.1. Incremental Exercise Test

PPO increased after training without differences between groups (SL: +20 ± 18 W, t = 3.09, CI = [4.58;34.42], *p* = 0.018, d = 0.95; 2000 m: +22 ± 19 W, t = 3.38, CI = [6.63;37.62], *p* = 0.012, d = 0.92; 3000 m: +16 ± 13 W, t = 3.4, CI = [4.55;28.02], *p* = 0.015, d = 0.94; 4000 m: +20 ± 11 W, t = 4.72, CI = [9.7;30.59], *p* = 0.003, d = 0.91) (Figure 1A). Regarding lactate, a main training effect for power at Dmax was found (*p* = 0.001, Figure 1B). Post-hoc analyses revealed that power at Dmax increased only at 2000 m (+14 ± 12 W, t = 3.29, CI = [3.91;23.9], *p* = 0.013, d = 0.94) and 4000 m (+12 ± 11 W, t = 2.74, CI = [1.28;22.54], *p* = 0.034, d = 0.93) (Figure 1B). The maximal blood lactate concentrations increased after training (main effect, *p* = 0.05) and the interaction between training and groups tended to be significant (main effect, *p* = 0.063) (Figure 1C). Post-hoc analyses revealed that maximal lactate concentrations increased after training at SL (+2.5 ± 2.4 mmol·L^−1^, t = 3.02, CI = [0.55;4.5], *p* = 0.019, d = 0.68) and tended to increase at 2000 m (+2.4 ± 3.6 mmol·L^−1^, t = 1.9, CI = [−0.6;5.45], *p* = 0.1, d = 0.67). Finally, no changes in maximal heart rate were detected in any conditions (Figure 1D).

### 3.2. Wingate Test

Mean power outputs measured during the Wingate test were higher after training (*p* < 0.001). Post-hoc analyses on the absolute values showed that mean power output increased at SL (+27 ± 31 W, t = 2.41, CI = [0.53;52.97], *p* = 0.047, d = 0.97), 2000 m (+21 ± 17 W, t = 3.47, CI = [6.77;35.74], *p* = 0.01, d = 0.97) and 4000 m (+31 ± 18 W, t = 4.46, CI = [14;48], *p* = 0.004, d = 0.94) but not at 3000 m (+20 ± 33 W, t = 1.57, CI = [−11.03;50.75], *p* = 0.17, d = 0.92) (Figure 2A). The peak power output was also improved by the training program (main effect, *p* = 0.001; Figure 2B). Post-hoc analyses showed that peak power significantly increased only at 4000 m (+38 ± 38 W, t = 2.63, CI = [6.62;72.52], *p* = 0.039, d = 0.94). A main training effect was found on lactate concentrations 3 min (*p* = 0.013, Figure 2C) and 5 min post-Wingate (*p* = 0.027, Figure 2D). Post-hoc analyses revealed that lactate concentrations tended to increase only at 3000 m 3 min post-Wingate (+4.2 ± 4.9 mmol·L^−1^, t = 2.23, CI = [−0.4;8.71], *p* = 0.067, d = 0.41) and at 2000 m (+2.5 ± 3.1 mmol·L^−1^, t = 2.25, CI = [−0.13;5.03], *p* = 0.059, d = 0.77) 5 min post-Wingate. Ultimately, the fatigue index, which represents the decrease in power during the test, tended to increase after training (main effect, *p* = 0.086, Figure 2E).

### 3.3. Time Trial Performance

Time trial performance (600 kJ) tended to improve after training (main effect, *p* = 0.092), but none of the individual group changes were significant (Figure 3A). No training effects were found in performance expressed in power developed during the time trial (Figure 3B) or in mean heart rate (Figure 3C).

### 3.4. Haematological Parameters

After training, hemoglobin mass (Figure 4A), hemoglobin concentrations (Figure 4B), hematocrit (Figure 4C) and plasma volume (Figure 4E) remained unchanged and no differences between groups were found. Red blood cell volume increased after the training program (main effect, *p* = 0.031) but none of the group increase was significant (Figure 4D). Ferritin concentrations decreased following the training period (main effect, *p* = 0.012). Post-hoc analyses showed that this decrease was only significant at SL (−20 ± 23 ng·mL^−1^, t = −2.38, CI = [−32.82;−0.09], *p* = 0.049, d = 1.05) and 4000 m (−16 ± 12 ng·mL^−1^, t = −3.33, CI = [−27.55;−4.22], *p* = 0.016, d = 1.28) (Figure 4F).

## 4. Discussion

### 4.1. Aerobic Fitness

After the training program, the aerobic fitness, assessed by PPO during an incremental exercise test, was increased in the same magnitude in each group, i.e., ~6%, indicating that a higher altitude does not elicit greater improvements. This observation is in line with previous reports showing a similar improvement in cycling aerobic fitness after SIH (F_i_O_2_ 14.4–15%, ~3000–2600 m) and SIN using 30 s all-out sprints [12,13,18,19]. Puype et al. showed a ~4% increase in time to exhaustion (TTE), reflecting PPO achieved during the incremental test, after 6 weeks (3 x/week) SIN or SIH [12]. In the present study, 2 sessions/week for the same period of time was enough to induce similar gains in PPO. A second study from this group showed that PPO and VO_2max_ were increased by about 10% after 5 weeks SIH (3 x/week) [18]. Of note, a smaller specific block of SIH (2 weeks, 3 x/week) had the potential to improve aerobic fitness; VO_2max_ was increased by slightly more than 10% [13,19] and PPO by 8% [13]. It is known that aerobic energy contribution can count up to 40% during a maximal 30 s effort, while shorter sprints require mainly anaerobic energy sources (ATP-PCr and glycolysis) [30,31]. Based on the latter observation and the literature available, aerobic adaptations such as increased PPO or VO_2max_ could be expected after SIH and SIN.

### 4.2. Lactate-Related Adaptations

During the incremental exercise test, lactate-related adaptations at submaximal intensity were exclusively observed in some of the altitude groups. The power output at Dmax, which is a mathematical approach to calculate lactate threshold, increased after training in our 2000 m and 4000 m groups but not 3000 m. This confirms previous work from Puype et al. (2013) showing an increase in power output at 4 mmol·L^−1^ lactate after 6 weeks with SIH only (~3000 m) [12]. However, despite similar training and hypoxic conditions to the above study, our 3000 m group did not significantly improve its power output at Dmax. This highlights that the different ways of determining the lactate threshold may lead to the distinct conclusions [32]. Taken together, these findings suggest that SIH may provide lactate-related adaptations that are not observed after SIN. Nevertheless, our observations do not identify an optimal altitude over another.

Considering these differentiated adaptations at submaximal intensity between SL and some of the hypoxic conditions, the question could be raised as to whether sport performance was affected. Despite a global trend to improvement, none of the four groups increased performance during the blind 600 kJ time trial after training. The absence of differences between SL and altitude groups is in line with the previously published SIH cycling studies that did not highlight different adaptations in comparison with SIN [12,13,18,19]. However, all previous studies reported an increased performance with both SIN and SIH [12,13,18,19]. Due to the disparity in the performance tests used, a comparison is difficult to make. Performance tests based on time (10 and 30 min time trials) [12,18] and submaximal TTE (80% of VO_2max_ or PPO, ~10 min) [13,19] have been used. To be close to the field reality of a time trial (be the fastest on a given distance), the 600 kJ test was chosen, during which the highest power needed to be produced to reach 600 kJ as quickly as possible. In this configuration, it seems that SIH-related increase in lactate threshold is not sufficient to enhance time trial performance. Similarly, in the study of Puype et al. (2013), the increase in power output at 4 mmol·L^−1^ lactate observed after SIH did not translate into superior performance gains in comparison to SIN [12]. Additional mental preparation and specific pacing training might be required to reach this purpose.

### 4.3. Anaerobic Component

The anaerobic component was tested using the Wingate test. An improvement in mean power was measured after training at SL, 2000 m, and 4000 m, with no difference between those groups. This result confirms previous reports showing that SIH and RSH do not provide any additional benefit over SIN and RST in normoxia (RSN) on the increase in mean power during a single maximal Wingate test [19,33]. Of note, two other studies failed to measure an improvement in Wingate mean power, suggesting that shorter concentrated blocks (2 weeks SIH [13] and 5 consecutive days RSH [34]) are not always sufficient to enhance anaerobic performance.

A main training effect was found for the Wingate peak power while the participants were trained at 4000 m. To our knowledge, sprint training studies using less severe hypoxic conditions (F_i_O_2_ 14–15%, 3200–2600 m) did not detect any change in peak power during a Wingate test either with short [33,34] or long sprints protocols [13]. However, Takei et al. (2020) observed an increase in peak power during repeated 30 s all-out efforts after two weeks (three sessions/week) of both SIH and SIN [35]. Additionally, peak power has been reported to be more improved during a repeated sprint ability (RSA) test after RSH than RSN [36]. After four weeks (two sessions/week, 7 s repeated sprints), the percentage of gain in peak power was superior in RSH compared to RSN [36]. The authors hypothesized that muscle phosphorylcreatine (PCr) content may be more increased in response to RSH, and in turn induce a greater improvement in peak power [36]. As expected from a biochemical point of view, this hypothesis was not supported by a second study from the same group looking at the effects of five consecutive days of either RSH or RSN. An increase in peak power output during the first sprint of the RSA test was observed only in the RSH group despite a similar increase in muscle PCr content between RSH and RSN [34]. It can therefore be hypothesized that the greater increase in maximum power output observed here during the Wingate test after training at 4000 m altitude (FiO_2_ 13.0%) is probably not related to energy metabolism, but to muscle structure or recruitment. Those conditions may have induced a higher expression of fast-twitch fibers [18,37] or increased the number of recruited motor units. As this study was not intended to highlight the mechanisms underlying the effects observed on sport performance, further studies will be useful to clarify them.

### 4.4. Haematological Parameters

As expected, only minor haematological changes were found. Hct and [Hb] did not change after the training period. This confirms previous findings showing no added value of SIH compared to SIN or any effect of training at all on these variables [13,19]. Hb_mass_ and PV did not change either. Although ferritin concentrations decreased following six weeks of training, mean values were well above the level of 20 ng·mL^−1^, generally used to indicate latent iron deficiency. A similar decline in ferritin concentrations has already been highlighted after a six-week HIIT program in cyclists [38]. In addition, increased iron losses through sweat, urine, exercise-induced hemolysis but also a reduced red blood cell lifespan is well documented in endurance athletes [14]. Despite the lack of change in Hb_mass_, the slight increase in RBCV observed following the training period could explain the decrease in ferritin concentrations as well. Of note, a similar magnitude of change was observed in ferritin concentrations at SL and 4000 m (19% vs 20%) suggesting that altitude does not increase the exercise-related iron loss as part of SIT.

### 4.5. Limitations

Some limitations must be considered regarding the results presented and discussed above. The main limitation of our experiment resides in the relatively low number of participants. Although, this number was sufficient to detect exercise-induced changes in PPO during the incremental test or mean and peak power during the Wingate test in specific groups, which was the purpose of the present study. Additionally, the effect sizes observed were mostly characterized as large. This said, caution should be taken to extend the results to a specific population, especially when considering interindividual heterogeneity of adaptations to exercise and hypoxia [39,40,41,42]. Secondly, our study can be defined as a single-blinded intervention, whereas a double-blinded design would have strengthened the study. Thirdly, nutrition was not controlled throughout the experiment, which represents another limitation knowing that nutrition is an important component of physiological adaptations to training. Ultimately, our study has focused on single aerobic and anaerobic performance tests but not repeated sprint tests. Further research could include an RSA cycling test to diversify the available results and open the practical perspectives.

### 4.6. Practical Applications

Major determinants of cycling performance are the PPO, the ability to utilize a large fraction of this PPO, and the cycling efficiency, but also the ability to suddenly develop very high power to perform breakaways, close gaps and win final sprints [43,44]. Based on our findings and the available literature, the use of SIH blocks seems to be an interesting strategy to enhance both aerobic and anaerobic capacity when performed for 6 weeks with 30 s all-out sprints. Specifically, our data show that the lactate threshold is improved after SIH compared to SIN. In addition, athletes should not hesitate to include a few sessions at an altitude as high as 4000 m to improve peak power. Taken together, those results show that SIH can improve most of the major determinants of cycling performance and some to a greater magnitude than SIN especially when performed at 4000 m. Finally, as previously mentioned, the results show great inter-individual variability in adaptation to training in hypoxia, implying that some athletes will benefit from a SIH block while others will not.

## 5. Conclusions

Both aerobic and anaerobic capacity were improved in a similar way in both SIN and SIH with no difference between altitudes, except for lactate-related adaptations, which were found only in SIH and Wingate peak power that only increased at 4000 m. Further research is needed to confirm this last finding and the underlying mechanisms, because this could be of particular interest for disciplines requiring the development of very high power such as in very explosive sports.

## Figures and Tables

**Figure 1 sports-08-00148-f001:**
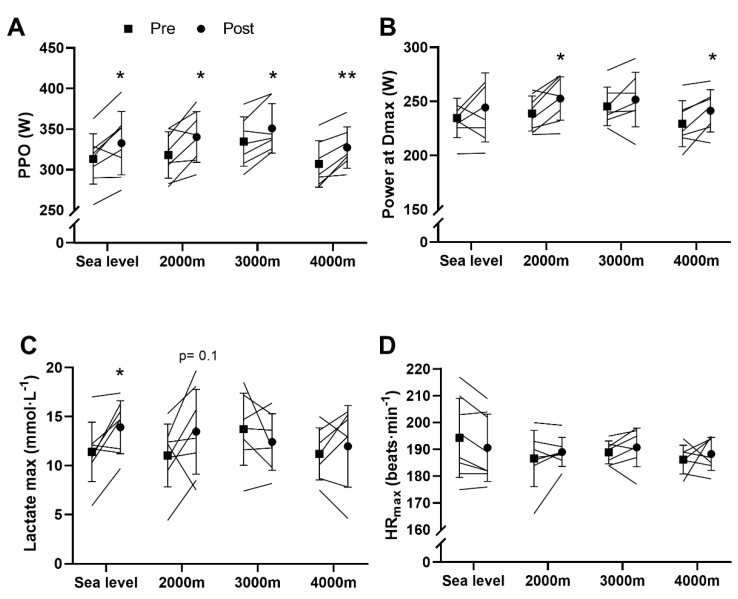
Incremental exercise test. (**A**) Peak power output (PPO); (**B**) Power at Dmax; (**C**) Maximal lactate concentrations; (**D**) Maximal heat rate (HR_max_). Values are means ± SD pre- and post-training. * *p* < 0.05; ** *p* < 0.01 vs pre-training.

**Figure 2 sports-08-00148-f002:**
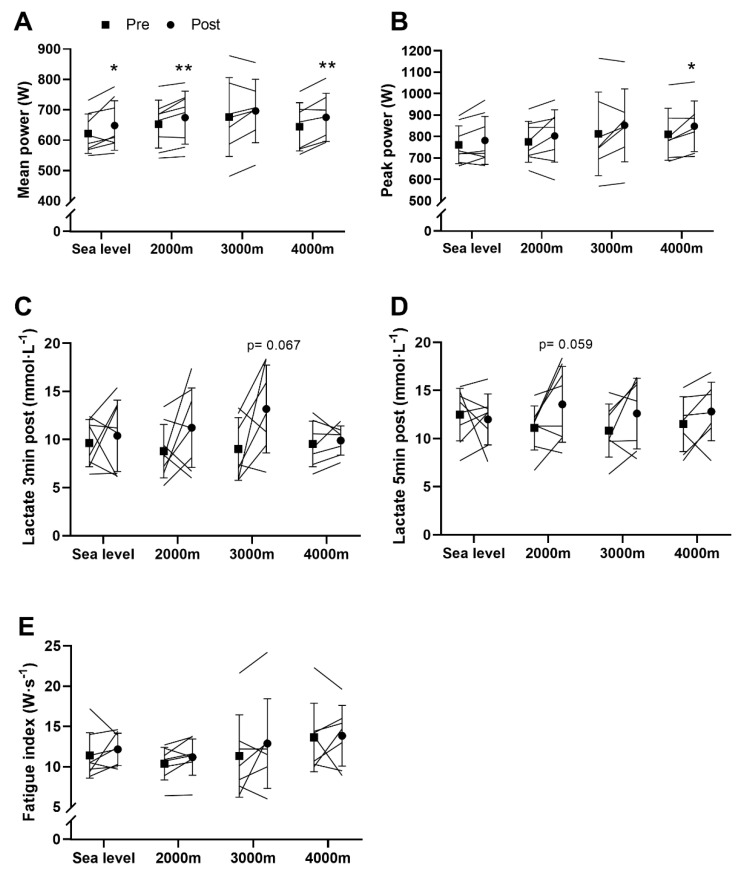
Wingate test. (**A**) Mean power; (**B**) Peak power; (**C**) Lactate concentrations 3 min post-Wingate; (**D**) Lactate concentrations 5 min post-Wingate; (**E**) Fatigue index. Values are means ± SD pre- and post-training. * *p* < 0.05; ** *p* < 0.01 vs pre-training.

**Figure 3 sports-08-00148-f003:**
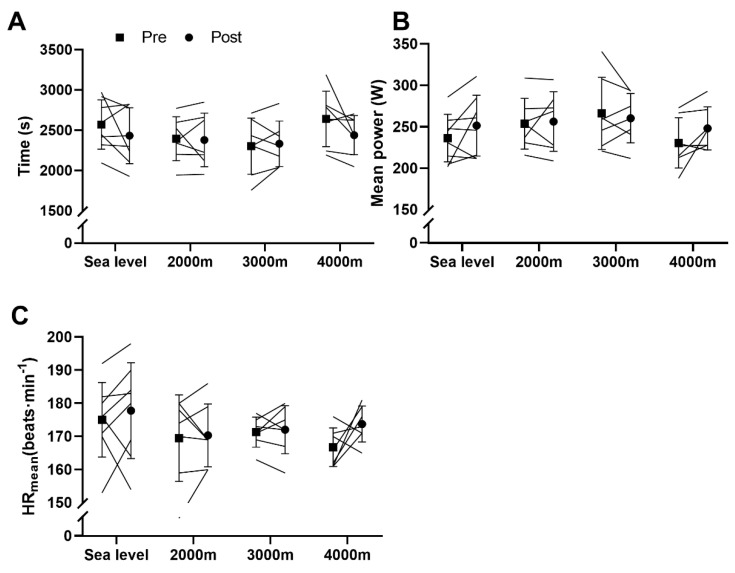
600 kJ time trial. (**A**) Performance in time; (**B**) Performance in power; (**C**) Mean heart rate (HR_mean_). Values are means ± SD pre- and post-training.

**Figure 4 sports-08-00148-f004:**
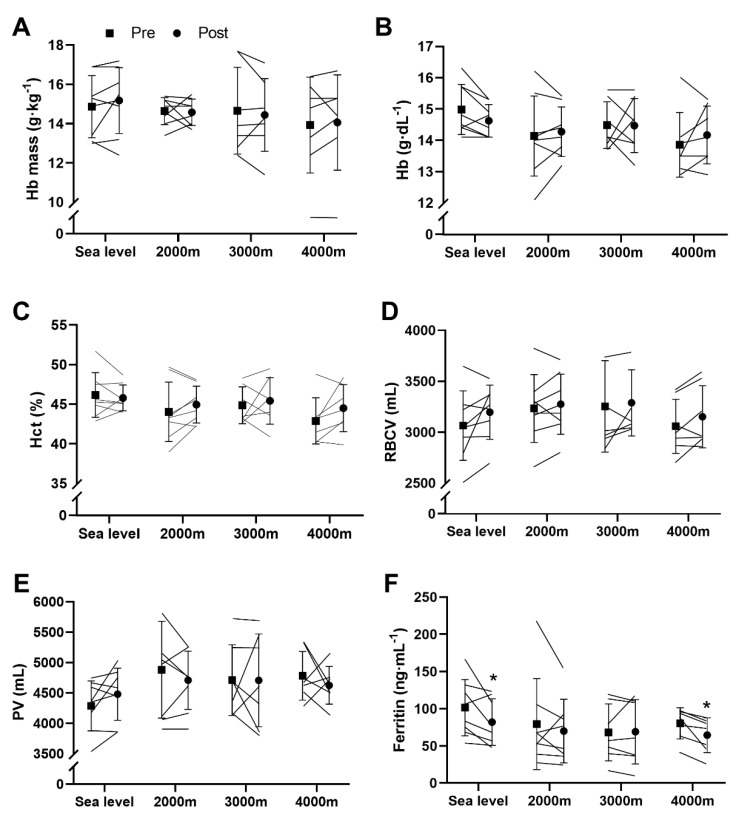
Haematology. (**A**) Relative Hb mass; (**B**) Hb concentrations; (**C**) Hematocrit; (**D**) Red blood cells volume (RBCV); (**E**) Plasma volume (PV); (**F**) Ferritin concentrations. Values are means ± SD pre- and post-training. * *p* < 0.05.

**Table 1 sports-08-00148-t001:** Participant characteristics.

	Sea-Level (*n* = 8)	2000 m (*n* = 8)	3000 m (*n* = 7)	4000 m (*n* = 7)
Age (year)	26.0 ± 4.4	25.4 ± 4.6	25.4 ± 4.2	25.1 ± 4.5
Weight (kg)	67.0 ± 4.1	71.2 ± 7.6	72.4 ± 9.9	72.6 ± 12.4
Basal PPO (W)	313 ± 31	318 ± 29	335 ± 31	307 ± 29

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
