# Peer review of "Effects of Sprint Interval Training at Different Altitudes on Cycling Performance at Sea-Level"

_sports, 2020, doi:10.3390/sports8110148_

Round 1
Reviewer 1 Report
I consider that your study is current and brings a substantial contribution to the knowledge of adaptations in various altitude conditions in cycling.
I would like to make the following recommendations:
-cycling should be introduced for keywords.
-in the protocol section to mention the date when the study was performed.
Reviewer 2 Report
General comments to the authors
The aim of the study was to determine if sprint interval training in hypoxia (SIH) at an altitude of 4000m can elicit greater adaptations than the same training at 2000m, 3000m or sea-level. The effect of SIT at different altitudes would be interesting to study in regard to aerobic vs anaerobic performance and not simply in terms of an “improved performance” or “greater adaptations” taken broadly. Moreover, authors went fishing for findings. Indeed, several performance-related parameters were measured, without providing clear apparent physiological reasons for justifying how altitude training may prove useful for each of them. This is not how science should be done.
Introduction
Clear apparent physiological reasons for justifying how altitude training may prove useful for each of the performance-related parameters studied must be provided, along, with their guidance, the awaited hypotheses on how they should be impacted.
Line 46 – “When performed in hypoxia, this type of training is defined as repeated sprint training in hypoxia (RSH) or sprint interval training in hypoxia (SIH).”
This sentence should be placed after the following one (after defining the difference between RST and SIT) to avoid confusion between these two types of training and highlight the fact that this study is about SIT.
Line 65 - “...optimal for maximizing the performance improvements induced by SIH programs.”
This is too vague as different training altitudes may have contrasting effect on aerobic vs anaerobic performance. To support this point, Kon, Nakagaki & Ebi (2019) showed that SIT under hyperoxia leads to a greater improvement in aerobic performance, but impaired anaerobic exercise performance.
Line 67 - The aims and hypothesizes of the study should clearly be stated in the introduction.
Materials & methods
The sample size was set to find a difference of 10%. The cited article (Puype et al., 2013) reports an improvement in performance of 6-8% compared to a non-exercising control group after 6 weeks of SIT. A 10% difference between training groups seems nearly impossible to achieve. The coefficients of variation reported for cycling incremental tests to measure peak power are usually 2% or less (Hopkins et al., 2001). Why was the difference set at 10%?
Line 96 – Please define what is semi-randomly?
Line 103 - Please report SD, not SEM.
Line 108 – that a familiarization trial was not performed before post testing represents a limitation of this study, if not a major flaw.
Line 121 – where was blood collected?
Line 132 – Please report the validity and reliability of this method?
Line 146 – Regarding the Wingate test, please report the fatigue index?
Line 158 – Data should be reported with the SD, not SEM?
Results
As the intervention was not tailored to induce any change in body mass, expressing the peak power output, wingate results or time-trial performances in relation to the body mass seems irrelevant. Especially if the intervention did not lead to any change in body mass.
As the Dmax method is more strongly correlated with performance (Heuberger et al., 2018), it is suggested to only present the results for Dmax and remove the power output at 4 mmol/L.
Discussion
Line 235 – Aerobic capacity is not representative of PPO
Line 235 – The improvement in PPO during a incremental exercise test may also be partly related to an improvement in anaerobic performance. Without the oxygen consumption value, it is hard to come to a strong conclusion in this regard. Especially since there were no improvements in 600 kJ TT and only some improvement in anaerobic performance.
Line 246 - Why is repeated sprints training discussed?
The study of Kon, Nakagaki & Ebi (2019) about SIT under hyperoxia could be relevant to discuss.
IMHO, the stated practical applications and conclusions seem a bit far-fetched in regard to the results of the study especially with a great discrepancy between individual responses.
Line 263 – lactate threshold or anaerobic threshold?
Reviewer 3 Report
Introduction
Although I know the objective, previously stated in the abstract, the authors should clearly state it in the introduction as it is already diluted with the hypothesis.
Use 3rd person and not 1st person throughout the manuscript.
Material and methods
What was the variable that allowed the participants to be randomized?
Substitute subjects for participants
Bearing in mind that nutrition is an important factor in exercise adaptations, the non-inclusion of these data is a very serious limitation that must be reflected. I know that the authors have indicated that the participants maintain their nutritional habits, but how did the authors control that this was the case? Are you sure that those of any of the groups did not take any nutrition that will allow them to obtain a better adaptation to hypoxia?
Why do the authors use SEM instead of SD? I think it is better to show the SD.
As for the statistics used, it seems to me that it is fantastic, perfect.
Did any of the participants take iron supplementation?
Conclusions
Delete the objective in the conclusions section and focus the conclusion on the objectives.
Reviewer 4 Report
Incredibly interesting, incredibly well done and detailed – I would suggest a little more background info on the subjects if possible – well done!
Line 43 – need ref
66-68 why was this your hypothesis?
Please give more subject info – were the competitive athletes? Recreation? A mix? Baseline performance would be helpful as well
Line 344 – I think you should change “potentiate” to “improve”
Round 2
Reviewer 2 Report
Authors must be congratulated for their efforts put into the improvement of the manuscript. However, I still have a minor comment to add regarding the conclusion.
Line 347 - It is written: “Finally, as previously mentioned, the results show great inter-individual variability in adaptation to training in hypoxia, implying that some athletes will benefit from SIH while others not. It is therefore important for an athlete to consider the fact that he could be part of the non-responders.”
The concept of ‘responders’ and ‘non-responders’ to altitude training likely goes beyond individual variations and the response to such training may be dependent on the interaction of multiple factors. Therefore, a 'non-responder' to a SIH block may be a 'responder' in a following SIH block. I suggest to simply remove the last sentence and rephrase the previous one:
“Finally, as previously mentioned, the results show great inter-individual variability in adaptation to training in hypoxia, implying that some athletes will benefit from a SIH block while others will not.”
Reviewer 3 Report
The authors have adequately worked on the manuscript. However, the nutritional aspects are not clear to me.
Since nutrition is a vehicle in physiological adaptations to training, I think this fact should at least be discussed in the limitations.
